



# Design Procedures and Experimental Verification of an Electro-Thermal De-Icing System for Wind Turbines

David Getz* and Jose Palacios†

*Pennsylvania State University, University Park, Pennsylvania 16802*

There has been a substantial growth in the total wind energy capacity worldwide. Icing difficulties have been encountered for cold climate locations. Rotor blade icing has been recognized as an issue and solutions to mitigate its effects have been identified. Wind turbines are adapting helicopter rotor and propeller ice protection approaches to reduce aerodynamic performance degradation related to ice formation. Electro-thermal heating is one of the main technologies used to protect rotors from ice accretion and it is one of the main technologies being considered to protect wind turbines. In this research, the design process required to develop an ice protection system for wind turbines is discussed. Three icing conditions were considered: Light, Medium and Severe. Light icing conditions were created using clouds at -8°C with a 0.2 g/m³ liquid water content (LWC) and water droplets of 20 μm median volumetric diameter (MVD). Medium icing condition clouds had a LWC of 0.4 g/m³ and 20 μm MVD, also at -8°C. Severe icing conditions had an LWC of 0.9 g/m³ and 35 μm MVD at -8°C. The design approach relies on modeling and experimental testing. Electro-thermal heater system testing was conducted at The Adverse Environment Rotor Test Stand at Penn State. Wind turbine representative airfoils protected with electro-thermal de-icing were tested at representative centrifugal loads and flow speeds. The wind turbine sections were ½ scale models of the 80% span region of a generic 1.5 MW wind turbine blade. A sample wind turbine configuration was selected to describe the design process. A 1.5 MW wind turbine was chosen. The icing cloud impact velocity was matched to that of a 1.5 MW wind turbine at full power production. Initially, ice accretion modeling was conducted to provide an initial prediction of the power density required to debond accreted ice at a set of icing conditions. Then, ice accretion thickness gradients along the span of the rotor blade for light, medium and severe icing conditions were collected experimentally. Given a pre-determined maximum power allocated for the de-icing system, heating zones were introduced along the span and the chord of the blade to provide the required power density needed to remove the accreted ice. The heating sequence for the zones started at the tip of the blade, to allow de-bonded ice to shed off along the span of the rotor blade. Given the continuity of the accreted ice along the bladed span, heating a zone could de-bond the ice over that specific zone, but the ice formation could not detach from the blade as it would be cohesively connected to the ice on the adjacent inboard zone. To prevent such cohesive retention of debonded ice, the research determined the minimum ice thickness required to shed the accreted ice mass with the given amount of power availability by not only melting the ice interface over the zone, but also creating sufficient tensile forces to break the cohesive ice forces between two adjacent heating zones. The quantified minimum ice thickness to overcome ice cohesive forces were obtained for all identified icing conditions. The experimental data was critical in the design of a time sequence controller that allows consecutive de-icing of heating zones along the span of the wind turbine blade. Based on the experimental and modeling efforts, de-icing a representative 1.5 MW wind turbine with a 100 kW power allocation required four sections along the span, with each heater section covering 17.8% span and delivering a 2.48 W/in² (0.385 W/cm²) power density. The time sequences for the controller required approximately 10 minutes for each cycle.

## Nomenclature

$A_{CS}$ – Cross-sectional area of accreted ice shape

$CF_{AERTS}$ – Centrifugal force experience in AERTS

$CF_{WT}$ – Centrifugal force experienced on the wind turbine

$L_{AERTS}$ – Span-wise length of the heater test section in AERTS

$L_{WT}$ – Span-wise length of the heater section on the wind turbine

$m_{AERTS}$ – Ice mass on the heater test section in AERTS

$m_{WT}$ – Ice mass on the heater section of the wind turbine

$\rho$ – Density of glazed ice in g/cm³

$\Omega_{AERTS}$ – Rotational velocity of the rotor blade in AERTS facility

$\Omega_{WT}$ – Rotational velocity on the wind turbine rotor blade

$r_{AERTS}$ – Radius from axis-of-rotation to center of heater section

$r_{WT}$ – Radius from axis-of-rotation to beginning of heater section

---

*Graduate Student, Research Assistant, Department of Aerospace Engineering, 207 Unit C Building.

†Professor, Department of Aerospace Engineering, 229 Hammond Building.



# I.    Introduction

Conventional energy like coal, natural gas and oil have gradually become a source of concern due to their environmental impacts. Wind energy has been used to generate electric power for over 100 years and the industry boomed during the oil crisis in the 1970s (Ackerman, 2002). In recent years, many developers are investing in wind farms in cold climate regions, because of the areas being remote, favorable wind conditions, and higher air densities. Among all the renewable sources, wind power has promising commercial prospects with the ability to produce large-scale electricity generation (Deal, 2010). Many countries have adopted this energy generation making wind market expand rapidly. Countries like China, Germany, United States, Denmark, Turkey and Spain have made substantial contributions towards the progression of wind energy (Hephasli, 2004, and Xu, 2010). In the last decade, the average annual growth for the world's wind power generation is approximately 30%. The global wind power capacity installed between 1990 and 2015 is shown in Figure 1. According to the World Energy Association, it is estimated that the capacity will reach 425 GW by 2015 (ABS Energy Report, 2010 and Leung, 2011).

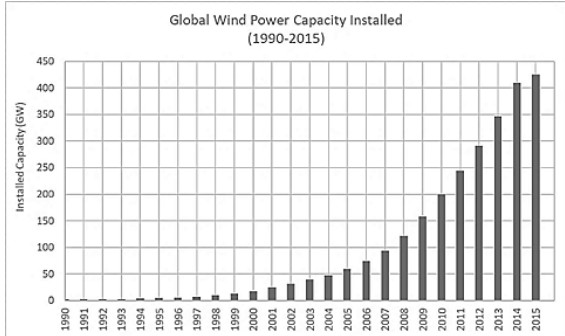

**Figure 1:** Global wind power capacity installed, GW, 1990-2015 (ABS Energy Report, 2010) .

In 2009, the U.S. had installed 10GW with a total installed capacity of 40.2GW. It is estimated that wind energy will generate 20% of the US electricity in 2030. Currently, it produces 2% of the nation's electricity (US Global Wind Energy Council, 2011). In 2010, China surpassed the U.S. as the world's leader of wind power. This is no surprise considering China's installed capacity doubled each year since 2006. The installed capacity of the leading wind power countries is demonstrated in Figure 2.

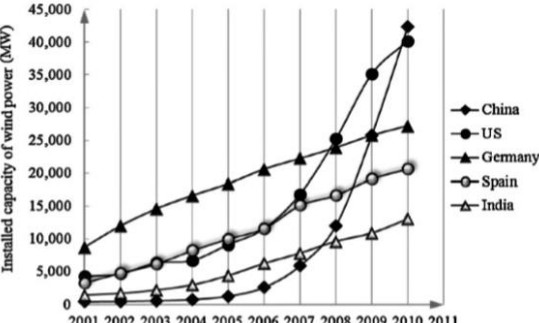

**Figure 2:** Installed wind power capacity in leading countries (US, PR China, Germany, India and Spain Global Wind Energy Council, 2011).

Most wind turbines experience hazardous icing events in cold climates. The accumulation of ice on the wind turbine blade modifies the blade geometry and degrade the aerodynamic performance. The estimated loss of the total annual energy production (AEP) can be approximately 20% Lamraoui, et al., 2014). Severe icing conditions cause rotational torque to decline to zero and the turbine is forced to shut down. Wind turbines also are stopped under icing conditions that cause heavy vibrations because these vibrations promote ice shedding, which can lead to imbalance forces that could damage the hub of the wind turbine and could pose threats to the surrounding environment due to ballistic shedding events (Lassko et al., 2003; Seifert et al., 2004 and 2003).

The ice accumulation problem is a complex phenomenon that combines various parameters. The shape of the ice and rate at which it accretes is dependent upon the atmospheric temperature, liquid water content of the cloud, impact velocity, water droplet size, and accretion time. The water content of the cloud is defined by the liquid water content (LWC) in $g/m^3$. The water droplet size in the cloud is characterized by the median volumetric diameter (MVD) in μm (Ruff et al., 1990). The Federal Aviation Administration (FAA) has defined two icing envelopes in the Federal Aviation Regulations Part 25 and Part 29 Appendix C using the MVD, LWC and ambient temperature. Continuous icing proves to be less severe, because the liquid water content ranges from 0.06 $g/m^3$ to 0.8 $g/m^3$ and the median value diameter of the droplets range from 10 μm to 40 μm. Typically, the LWC and MVD affect the thickness of the ice shape, while temperature and droplet impact velocity affect the surface roughness and adhesion strength of the ice. Icing envelopes for wind turbine regions have not been officially established.





Icing conditions on wind turbines have led to research in various anti-icing and de-icing ice protection systems (IPS). Anti-icing systems
simply keep the rotor blades free from ice. Anti-icing systems consume massive amounts of power because they operate continuously and must protect
large areas to prevent risks of water re-freezing in aft locations. De-icing systems allow thin layers of ice to accrete and then rely on centrifugal forces
to assist with ice shedding. De-icing systems are less power expensive than anti-icing systems, because they only have to protect the ice accretion
region within ice impingement limits. Also, since accreted ice acts as an insulator, the de-icing system is not subjected to the levels of convective
cooling that an anti-icing system would encounter. De-icing systems attempt to minimize runback water from freezing on the aft section of the blade.
The re-freezing of runback water can be prevented by evaporating the ice interface created by the impingement of super cooled water droplets. In
general, the evaporative mode for anti-icing systems require about 5 times more energy to operate, rendering it as too expensive for wind turbines.
The wind turbine industry has borrowed ice mitigation techniques from the aviation industry. In addition to electro-thermal de-icing, the
aviation industry has used active ice protection systems and is working on developing passive techniques. *Passive* ice protection systems consist of
coatings that are characterized by low ice adhesion strength. Passive ice-phobic coatings degrade under erosion conditions, which cause them to lose
their ice-phobic properties. To date, a fully passive ice protection coating with application to the aviation industry does not exist. Some examples of
active ice protection systems are hot-air injection, ultrasonic waves, electro-thermal heaters and pneumatic boots (Oermeyer et al., 2012; Martin et al.,
1992; Flemming, 2003; Botura et al., 2005; Buschhorn et al., 2013).
Electro-thermal ice protection systems are the most common IPSs used due to the simplicity. The system sends electrical current to resistive
circuits, which convert electrical energy into thermal energy. These resistive circuits are known as the heating elements. These heating elements are
typically adhered underneath the blade skin and coverlay. A simplified schematic of the layers is displayed in Figure 3. Coverlay is a material laminated
to insulate the copper conductor. The blade skin is the layer of protection used to prevent surface erosion. The thermal energy converted from electrical
energy in the heating elements travel through the layers via conduction.

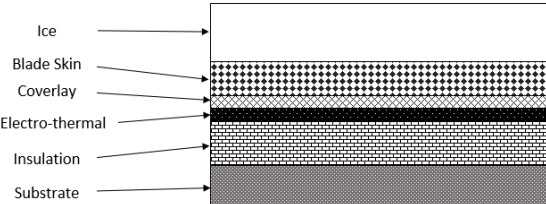

**Figure 3:** Simplified electro-thermal IPS schematic.
With latitude coordinates like Canada, electro-thermal ice protection systems are used in Pori and Olostrunturi, Finland. Between 1999 and
2001, Pori's IPS was mainly used for public safety and its power consumption was 1% of the annual production. In extreme events, the IPS used a
maximum of 6% nominal power. In Olostrunturi, the IPS used 3.6% of the annual production (Lasko et al., 2003 and 2005).
This paper concentrates on the design process of an electro-thermal de-icing heater configuration for wind turbines. The conducted research
relies on experimental data obtained for an electro-thermal de-icing IPS to determine heater zones and required ice accretion times to promote ice
shedding with assistance of centrifugal loads action on the accreted ice.

## II.    Objectives

This research presents the design process of an electro-thermal ice protection systems (IPS) for wind turbines. The design process begins with
predicted values of power density requirements and then relies on experimental results to verify and quantify critical parameters such as ice accretion
rates, minimum power densities and required loads to promote cohesive failure between adjacent zones. Representative icing conditions were selected
and guided by FAR Appendix C icing conditions typical of aircraft environments. The goal of the ice protection research for wind turbines is to develop
methodology to design such de-icing systems. A generic 1.5 MW wind turbine is used to describe the design process. Power limitations on the wind
turbine (100 kW) were set as a priority. Shedding times were also enforced (30 seconds from the initialization of the heaters).
Firstly, the 2D heater configuration was modeled in LEWICE, a NASA developed ice accretion prediction software[31]. The modeling results are
used to predict ice accretion for varying power densities to the electro-thermal heaters. Secondly, the ice accretion thickness rate must be verified for
several icing conditions, as well as the minimum ice thickness needed to promote effective ice shedding and overcome cohesive bonding between span-
wise zones. Lastly, power density variation must be explored to quantify the change in shedding times and to enforce the maximum shed time of 30
seconds. To meet this objective, a time sequence controller was designed for the selected icing regimes (light, medium, severe). The conducted research
introduces designed guidelines, development and testing approaches for an electro-thermal de-icing IPS for wind turbines.

## III.    Experimental Configuration

### A.    Facility Overview

The Adverse Environment Rotor Test Stand (AERTS) was designed and constructed at the Vertical Lift Research Center of Excellence at
The Pennsylvania State University (Brouwers et al., 2010). A photograph of the facility equipped with NACA 0012 test blades is exhibited in Figure
4. This photograph shows the QH-50 DASH Hub, 125 HP motor with a built-in torque sensor, Bell Housing with 6-axis load cell, and the
collective/cyclic pitch actuators. The hub was donated by Gyrodyne Helicopter Historical Foundation. The QH-50D is a co-axial UAV designed for
the Navy in the early 1950's (Yan, 2016). The test stand is surrounded by an octagonal ballistic wall. The surrounding walls of the freezer have
dimensions: 6m x 6m x 3.5m.





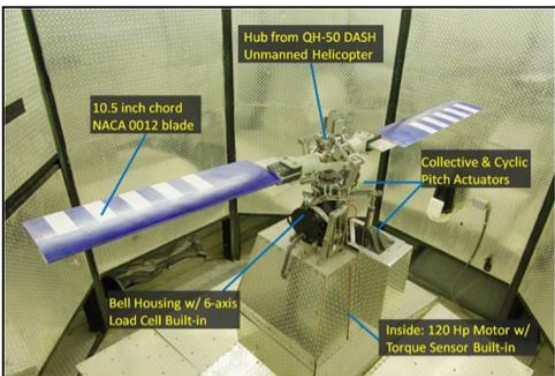

**Figure 4:** Photograph of AERTS test stand.

The chamber is cooled by a 10 HP compressor/evaporator system with controllable temperatures between -25°C and 0°C. The rotor can achieve
RPMs up to 1600. The 6-axis load cell measures forces and moments, while the torque sensor monitors the torque provided from the motor. The
standard icing nozzles are donated by the NASA IRT and are installed on the ceiling of the chamber. The nozzles are arranged in two concentric rings
containing 5 nozzles on the inner ring and 10 nozzles on the outer ring. The nozzle system operates by aerosolizing water droplets with a combination
of air and water at precise pressures (Ide et al., 2001). The nozzle calibration curve is presented in Figure 5, which depicts the MVD of the droplet on
the y-axis in microns and air pressure differential on the x-axis in pounds per square inch.

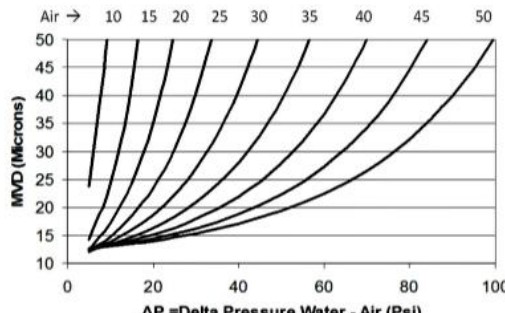

**Figure 5:** NASA standard icing nozzle operation chart (Ide et al., 2001) .

**B.         Blade Configuration**
The de-icing research conducted for the wind turbine needs to have a testing blade that is representative. The airfoil at the tip of the rotor blade
tested has a 72.4 cm (28.5 in) chord with a span of 30.48 cm (12 in). The DU 93-W-210 airfoil with the carrier blade amounts to a 139.7 cm (55 in)
total span to the rotation axis. The physical dimensions of the rotor test blade are illustrated in Figure 6 (Blasco, 2015 and Han 2011). All rotor blade
tests were conducted at 0° angle of attack (AoA). Figure 7 is a photograph of the test blade. The leading edge of the heating elements are protected by
an erosion tape with thermistors underneath. The span width of the heaters is slightly oversized in efforts to prevent ice bridging.

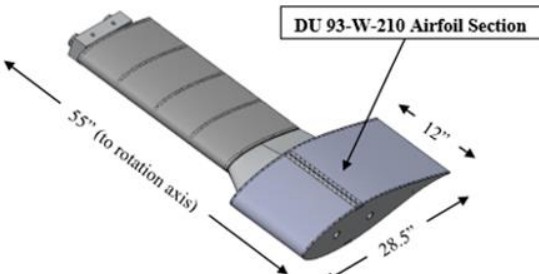

**Figure 6:** Wind Turbine test blade representative (Han, 2015).

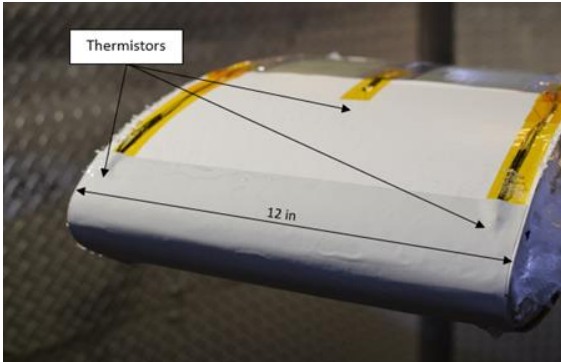

**Figure 7:** Rotor test blade.

To represent the centrifugal forces on the accreted ice, representative ice shapes were accreted and the RPM of the AERTS truncated rotor was
adjusted to match the centrifugal loads of the full-scale wind turbine at varying span locations. A cloud density of 0.917 g/cm$^3$ was selected for the
testing. The density value was selected as it is representative of that seeing on wind turbines and since it is a typical glaze ice condition that can be
reproduced in other facilities, such as the NASA Glenn Icing Research Tunnel (Vargas et al., 2005).
The cross-sectional area of the ice shape was photographed after each test. A metal hot plate was used to smoothly cut any three dimensional
ice features at the tip of the blade. A perpendicular plane against the 2D ice shape was obtained. The photograph contained a scale used to digitize the
image (Schneeberger, 2016). The estimated cross-sectional area of the accreted ice was quantified. An example of this digitalization approach is shown
in Figure 8.

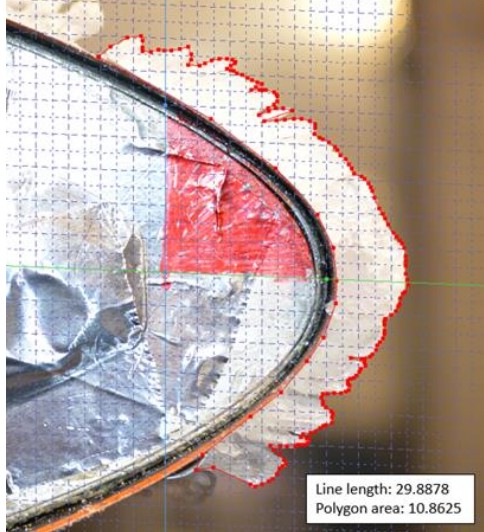

Line length: 29.8878
Polygon area: 10.8625

**Figure 8:** Digitized ice shape at -8°C with 0.4 g/m$^3$ LWC.

In this picture, the *x* and *y* axes (legs of red right triangle) are 3 cm in length. The estimated ice area of the polygon for the specific test shown
is 10.86 cm$^2$ and the perimeter of the selected 2D ice area is 29.89 cm. Given density and the length of the protected airfoil (30.48 cm), the estimated
ice mass is 303.54 grams. The assumed ice masses were calculated for these experiments as per Equations (1) and (2).
$$m_{WT} = A_{CS} * L_{WT} * \rho \qquad (1)$$

$$m_{AERTS} = A_{CS} * L_{AERTS} * \rho \qquad (2)$$

The centrifugal forces on the wind turbine and the AERTS facility were equated for a given span location on the full-scale blade. Then the
appropriate rotational velocity parameter for the AERTS facility was obtained. Equations (3-5) summarize the process.



$$CF_{WT} = m_{WT} * \Omega_{WT}^2 * r_{WT} \qquad (3)$$

$$CF_{AERTS} = m_{AERTS} * \Omega_{AERTS}^2 * r_{AERTS} \qquad (4)$$

$$\Omega_{AERTS} = \sqrt{\frac{CF_{WT}}{m_{AERTS} * r_{AERTS}}} \qquad (5)$$

The IPS heater configuration must be designed following operator requirements and restrictions. Assumptions were made in the design effort
presented in this paper. The maximum allowed power for the IPS is set to 100 kW. As it will be shown, the power limitation makes it impossible to
protect the entire leading-edge surface of the blades, forcing to partition the de-icing system to ensure that sufficient power densities are available.
Based on initial LEWICE models, each turbine blade had to have 4 span-wise heater sections beginning at 26.7% span from the root of the rotor blade
and ending at 97.7% span. The span percentages chosen for experimental representation correspond to the beginning of each heater section. A schematic
of the heater zones is illustrated in Figure 9. These span partitions allow cohesive bonding of ice between zones to play a role in the ice shedding events.
Before the ice accretion reaches a critical thickness, the cohesion strength of the ice exceeds the centrifugal forces to shed the ice layer. Therefore, a
minimal ice thickness (mass to be exact as it is linearly related to the load) exist to promote ice shedding (minimum ice thickness such that centrifugal
loads are enough to exceed the cohesion strength between zones). Selecting the inner-most span location of each heater section as the point to calculate
centrifugal loads for an ice shape is a conservative approach to ensure that the force generated at the inner location of the heater zone is enough to
overcome cohesive forces. The load acting on the accreted ice is a distributed load that increases linearly span-wise. A schematic of the cohesive force
opposing the centrifugal force is shown in Figure 10.

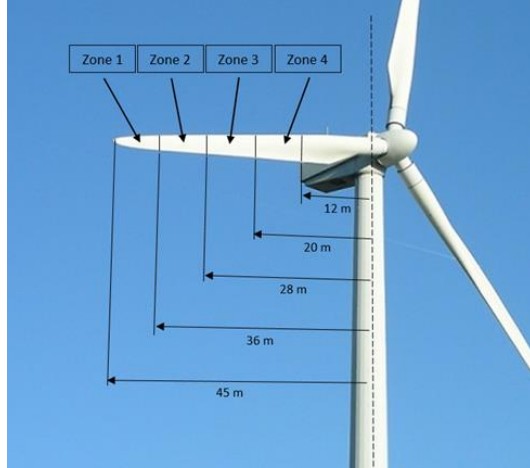

**Figure 9:** Schematic of full-scale wind turbine

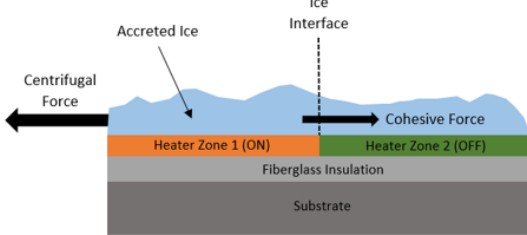

**Figure 10:** Schematic of cohesive force opposing the centrifugal force.

## IV.     De-Icing IPS Design Overview

Many parameters must be considered during the design phase of an ice protection system. A flowchart summarizing the design process for a
de-icing ice protection system is shown in Figure 11. Utilizing an analytical heat transfer modeling software allowed establishing an initial power
density requirement. Based on this requirement, partition zones can be identified. The dimensions of the heater zones must be selected and modeled
based on the maximum power availability reserved for the heating system. The heater configuration can be optimized by performing iterations in the
modeling phase to find the lowest initial power density. This power density is then used in experimental efforts to find the minimum ice thickness
needed for effective shedding, including the effects of zone-to-zone cohesion, for a desired heating time (less than 30 seconds is used in this document).



The testing phase is categorized by "Rotor Environment" and "Power Variation" experiments. To conduct these tests, the testing temperature
in the cooling chamber is selected first. Rotor Environment testing begins by matching the droplet impact velocity along the span of the rotor blade.
These initial tests identify the ice accretion rate along the span of the rotor blade for various LWC values at representative droplet impact velocities.
The environment temperature provides knowledge of expected accretion rates for a given icing condition. Temperature and LWC inputs to the
controller are used to select the appropriate time sequence to the heaters.
Once representative ice has formed, the centrifugal force acting on the ice shape must match the centrifugal force of the full-scale rotor blade
span at each heater zone location. By matching ice loads, these experiments will investigate the boundary of cohesive failure between heater zones. If
the ice thickness is too small, the cohesive force of the ice will dominate centrifugal forces, even if the ice-surface interface on the airfoil is melted.
Therefore, there exists a minimum centrifugal force required to effectively shed the ice layer from the rotor blade (or ice mass for a given RPM).
During these tests, ice is accreted to a desired thickness. Then the rotor stand spins to the appropriate RPM that matches the full-scale centrifugal
load of the inner-zone location of the heater and delivers the selected power density to the heater obtained in the modeling phase. If ice layer sheds in
less than the allocated heating time (30 seconds), the next test will decrease the accretion time, which decreases the ice mass. The objective is to find
the minimum ice thickness needed for "effective" shedding at each heater zone location. After the minimum thicknesses are found, the 2D cross-
sectional area of ice is digitized to obtain the ice mass and calculate the centrifugal force.
The final set of experiments introduce power variation to the heaters. These experiments quantify the time of the shedding events as power
density is decreased.
Lastly, the information gathered through this design process produces the capability to design time sequences for each icing condition explored.
The time sequences are used by the controller after ice is detected at a given icing condition. As an example of implementation, once ice is detected by
ice accretion sensors, a timer begins, and the ice thickness detector will read a thickness after one minute (Liu, 2017). The ice accretion rate will
determine the icing condition and the appropriate heater sequence is activated. Details on the steps taken to design the heater controller sequence are
described in detail in the next sections.

## V.  De-Icing IPS Experiments

### A.  Linear Ice Accretion Thickness

The objective of these tests was to find the ice accretion rate (slope of ice thickness over time) along the span of the rotor blade at three different
icing conditions. Data from these experiments play a role in the design of the de-icing time sequence controller. A Light icing condition was triggered
by a 0.2 g/m3 LWC and 20 μm MVD at a temperature of -8°C. A Medium icing condition consisted of a 0.4 g/m3 LWC and 20 μm MVD at -8°C.
Lastly, 0.9 g/m3 and 35 μm MVD combination was selected for the Severe icing condition. The nomenclature for the icing conditions are used
throughout the de-icing experiments and results.
The RPM used in AERTS during ice accretion must represent the water droplet impact velocity on the wind turbine. For this study, convective
cooling plays a major role in the initial stages of ice accretion. The RPM correlation in AERTS that represents the impact velocity at a chosen span
percentage of a wind turbine was therefore calculated. After spinning up to the appropriate RPM, the icing cloud was turned on for three different
durations: 3, 5 and 7 minutes. The three icing durations would experience the light, medium and severe icing conditions. The key parameters for this
set of experiments is shown in Table 1.
**Table 1:** Linear ice accretion test matrix.

| Span % | Impact Velocity (m/s) | AERTS RPM | Accretion Time (mins) |
|---|---|---|---|
| 26.7 | 22.65 | 174 | 3, 5 and 7 |
| 44.4 | 37.66 | 289 | 3, 5 and 7 |
| 62.2 | 52.76 | 405 | 3, 5 and 7 |
| 80.0 | 67.86 | 521 | 3, 5 and 7 |


Unfortunately, 80% span was not tested in the AERTS facility due to the potential rotor imbalance on the test stand at such RPM. At 67.86 m/s
tip speeds (521 RPM), both blades would need to shed simultaneously, or the force imbalance would become too large for the rotor stand. Note the
impact velocity is linear with RPM, while the centrifugal loads are square of the RPM. Therefore, once the desired ice shape was accreted, the RPM
was reduced to match centrifugal loads of the full-scale span location.
The data produced from this research, as expected, found a linear ice accretion along the span of the rotor blade. If the onset of the ice accretion
is known, then the accretion rate delivers the capability to know the thickness at each span-wise heater location at any future time. The ice thickness at
span percentages outside of the range, 26.7% to 62.2%, can be extrapolated from the data trend. The ice accretion for 3, 5 and 7 minutes at *light* icing
conditions is shown in Figure 12. The ice thickness for *medium* and *severe* icing conditions are presented in Figures 13 and 14, respectively.
The data has strong positive correlation and demonstrates a linear behavior for ice thickness along the span of the rotor blade. The experimental
measurements can now be used to obtain the ice accretion rate at each zone location.
Ice detection hardware would be required to identify the initial presence of ice on the rotor blades. After a selected time period, the ice detection
sensor can measure the ice thickness and the controller interface can calculate the ice accretion rate (Liu, 2017). This ice accretion rate determines the



severity of the icing conditions and activates the appropriate heater sequence. The accretion rates for each zone in all the icing conditions is presented
in Table 2. Interpolation between these icing conditions is possible.

**Figure 11:** Flowchart identifying the design of the de-icing IPS procedure.






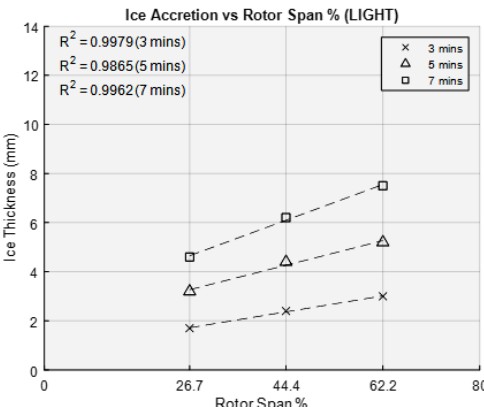

**Figure 12:** Ice thickness measurements for Light icing conditions.

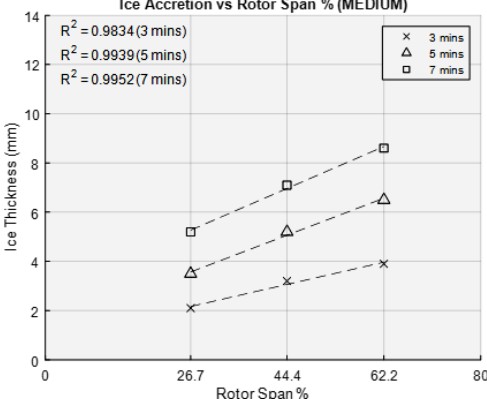

**Figure 13:** Ice thickness measurements for Medium icing conditions.

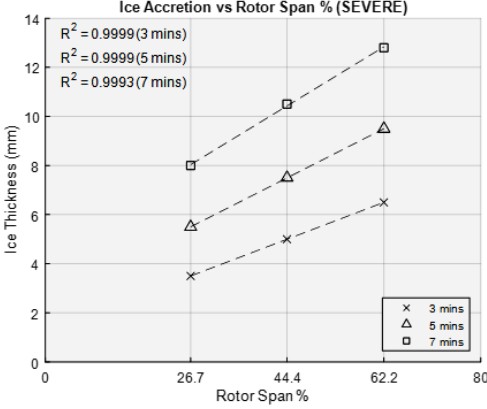

**Figure 14:** Ice thickness measurements for Severe icing condition.




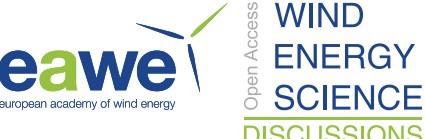


**Table 2:** Ice accretion rates (mm/min) for each heater zone.

| Heater # | Span % | Accretion Rate (LIGHT) | Accretion Rate (MEDIUM) | Accretion Rate (SEVERE) |
|---|---|---|---|---|
| 1 | 26.7 | 1.350 mm/min | 1.450 mm/min | 1.850 mm/min |
| 2 | 44.4 | 1.125 mm/min | 1.175 mm/min | 1.575 mm/min |
| 3 | 62.2 | 0.950 mm/min | 0.975 mm/min | 1.375 mm/min |
| 4 | 80.0 | 0.725 mm/min | 0.775 mm/min | 1.125 mm/min |


**B.   Minimum Ice Thickness for Shedding**
It is critical that the ice mass sheds in a timely manner before it becomes a hazard to the infrastructure of the wind turbine or the surroundings.
Also, the allowed accreted ice mass must not produce high aerodynamic performance degradation. The following test cases identify the minimum ice
thicknesses needed to promote ice shedding at each heater section. The span percentage at the origin of each heater section was selected to calculate
the effective centrifugal force acting on the accreted ice. Cohesive bridging was investigated as the ice mass decreased. For shedding to occur, once
the ice has debonded from the surface due to heating, the adhesion strength of the ice bridging surface between zones must be less than the centrifugal
force pulling on the ice mass in the span-wise direction. If the ice mass is too small, the tensile adhesion strength will dominate over the centrifugal
force, creating the inability to shed ice as it will remain attached to the most inner ice section that has not been debonded. Table 3 presents the parameters
for these experiments.

**Table 3:** Properties for Minimum Ice Thickness Shedding experiments.

| Heater # | Span % | Shedding RPM in AERTS | Heater Power Density (W/cm²) | Droplet Impact Velocity (m/s) |
|---|---|---|---|---|
| 1 | 26.7 | 286 | | |
| 2 | 44.4 | 369 | 0.385 | 52 |
| 3 | 62.2 | 437 | | |
| 4 | 80.0 | 496 | | |


Each test case accreted ice at -8°C and an impact velocity of 52 m/s (400 RPM in AERTS), deeming the ice accretion rate constant through
each icing condition. The impact velocity mostly affects the ice shape and has small effects on the heat transfer once ice is accreted (ice acts as an
insulator). The increase of velocity for inner sections with respect to what is expected on a full-scale blade (26.7% and 44.4%) simply means that the
accretion rate was increased (since the ice accumulation parameter is linearly dependent on LWC and time (Anderson, 1994 and Rocco, 2016), and
does not have any effects on the capability of the heaters to melt the ice interface. It must be noted that the ice shape could deviate slightly from those
accreted at representative span velocities, but ultimately, only the ice mass is of importance for this study. After the test blade accreted an ice shape for
a predetermined time, a hot plate was used to remove the tip ice shapes of the paddle blade to create a smooth plane on the edge. Photographs of the
cross-sectional area of the ice shape were obtained. After the photograph was taken, the ice on the outer edge of the heater at the tip was removed to
ensure the ice mass was free to shed. This represented the outboard heater section on the full-scale turbine that was free of ice. This step had to be done
due to the lack of heat production above the outer copper bus bar in the heating element. The copper bus bars, lacking heat generation, were embedded
in the heater layer and reside on the inner and outer edge of the heater section. Therefore, the inner section of the heater represented the adjacent inboard
heater section on the full-scale wind turbine that has no heat generation. This procedure was repeated for the *light, medium and severe* icing conditions.
Figure 15 is a front-view schematic that illustrates the cohesive attachment of the inboard section representative of the full-scale turbine. This procedure
experimentally evaluated the cohesive forces of the ice bridging the heater sections. Figures 16 and 17 show the procedure for ice removal on the outer
edge of the heater section representing the outboard heater section free of ice on the full-scale turbine.

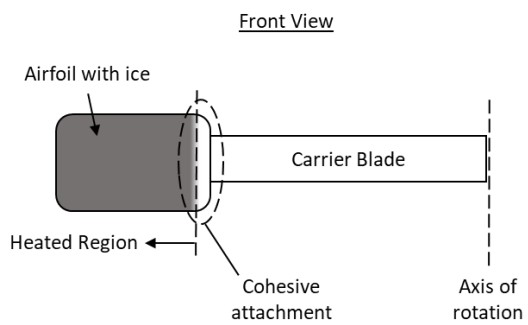

**Figure 15:** Schematic of representative inboard heater section with no heat.




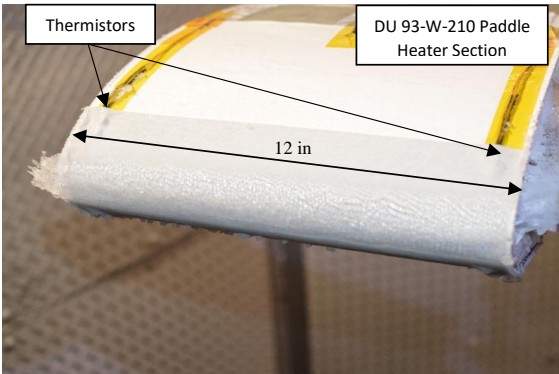

**Figure 16:** Heater section with 4 mm ice accretion before ice removal.

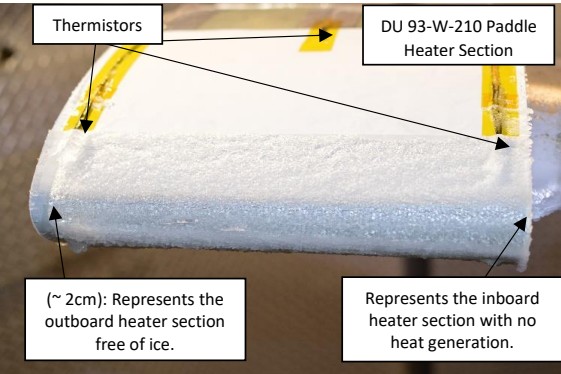

**Figure 17:** Photograph of the ice removal procedure.
After the rotor test blade is prepared for shedding, the rotor stand spun up to the appropriate centrifugal force representative of the full-scale
blade span percentage of interest. When the rotor stand reaches the desired RPM, the heaters were activated and delivered a 0.385 W/cm$^2$ power density,
which is the maximum available due to power restrictions on the full-scale wind turbine (corresponding to 100 kW and a partition of 4 zones per blade).
The time limit for effective ice shedding was set to 30 seconds. If the ice mass shed within 30 seconds, the following test case will accrete ice for a
shorter duration of time, decreasing the thickness/mass of the ice. This process was conducted until the minimum ice thickness was found for each
heater section. After the test was completed and the data was saved, the rotor blades were cleaned in preparation for the next test with reduced ice mass.
An example of a 7 mm ice shape at -8°C obtained in the medium icing conditions prepared for shedding is shown in Figure 18. Figure 19 is a photograph
of a successful shedding event after 23 seconds for the 26.7% span heater zone (innermost heater zone on the full-scale wind turbine). The data trends
for these tests are presented in Figure 20.

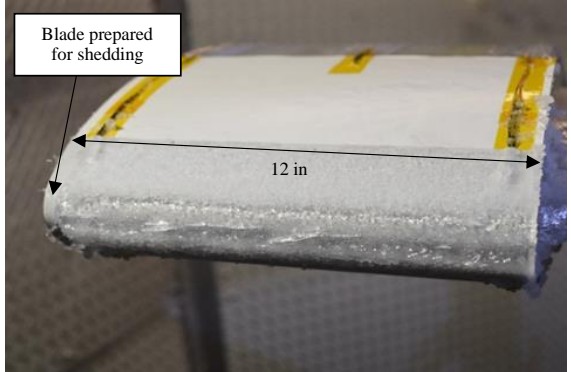

**Figure 18:** 7 mm ice shape at -8°C in Medium icing conditions.





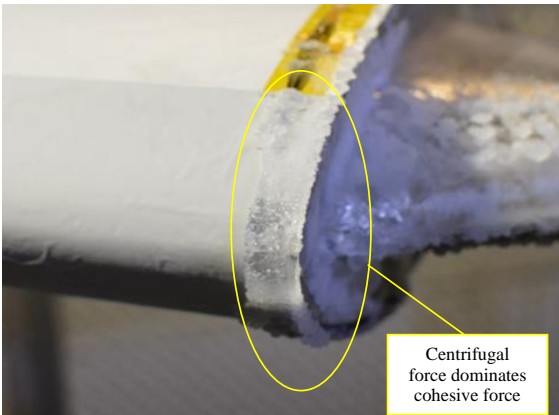

**Figure 19:** Successful shedding event overcoming ice cohesive forces.
The minimum ice thicknesses experimentally determined for the *light* icing conditions at the 26.7%, 44.4% and 62.2% span were 6 mm, 4 mm
and 2.8 mm, respectively. The average shedding time between both rotor-blades for these tests were 24, 25 and 28 seconds. For the *medium* icing
conditions, the minimum ice thicknesses were 3.7 mm, 5 mm and 6.8 mm, respectively. The average shedding time for these cases were 25, 25 and 23
seconds. Lastly, the minimum ice thicknesses for the *severe* icing conditions were 4 mm, 5 mm and 7.2 mm, respectively. The ice shedding times for
these tests were 30, 24 and 20 seconds.

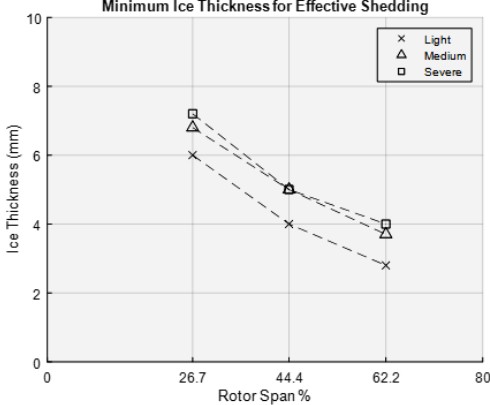

**Figure 20:** Minimum ice thicknesses needed for effective shedding.
The minimum thicknesses along the span of the wind turbine blade for the *severe* icing conditions should be considered for the design of the
controller. This is a conservative approach, because the smallest variation of ice mass could allow the ice tensile adhesion strength to dominate over
the centrifugal forces. In this situation, the ice shape stays attached to the airfoil as the electro-thermal heaters melt the ice interface on the surface. The
ice becomes an insulator and continues to accrete ice on the leading edge. It is imperious to keep these ice thicknesses at a minimum for the safety of
the surrounding environment on the wind farms.

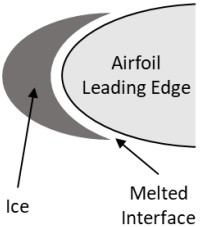






**Figure 21:** Schematic of melted interface experience from cohesive bonding.
This cohesive "holding" event of a debonded ice zone happened multiple times during the experiments. The following photographs are from a
test case demonstrating the failure to shed and the success of the ice tensile adhesion strength over the centrifugal forces. A side-view schematic from
the tip of the melted interface experienced from cohesive bonding is shown in Figure 21. The melted interface after delivering a 0.385 W/cm$^2$ power
density from the heaters for 180 seconds and failing to shed is shown in Figure 22. This is an example of the ice mass not reaching the critical value to
successful produce cohesive failure. As stated earlier, on the inboard section of the heater zone, there is a lack of heat generation on purpose to simulate
the cohesive forces in the testing environment. This represents full-scale bridging effects from zone to zone in the field.

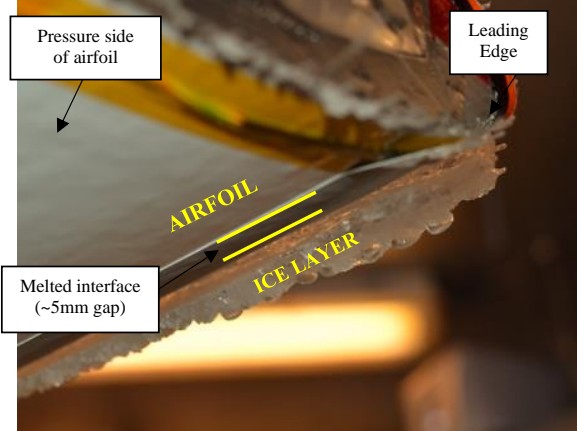

**Figure 22:** Melted interface between ice and airfoil leading to cohesive bonding and failure to shed.
The minimum ice thicknesses along the span for the severe icing condition were used to determine the cohesive failure curve. The cross-
sectional area of the ice shape was photographed and digitized to obtain the 2D area, as stated earlier. The ice mass was used to calculate the centrifugal
force at each heater span location. This data curve represents the minimum centrifugal force needed to exceed the ice cohesive force. If the ice thickness
(mass) exceeds the boundary, centrifugal forces will dominate the cohesive forces. The boundary curve is presented in Figure 23.

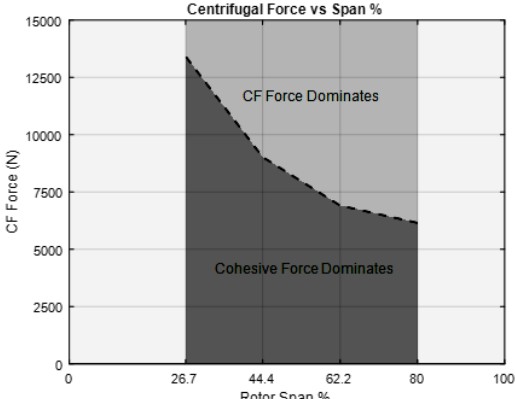

**Figure 23:** Cohesive failure curve.
**C.**     **Power Density Variation**
The last set of experimental tests was conducted to determine the minimum power density needed to debond the accreted ice within the allocated
heating time. Variations of power density delivered to the electro-thermal heater zones was investigated. These experiments were performed for the
minimum thickness needed to prevent cohesion between icing zones and was analyzed for each span-wise heater zone. The objective was to quantify
the required shedding times for reduced power densities at the found minimum ice thickness. As stated in the previous section, effective ice shedding
means the shedding event occurred within 30 seconds. Each test case accreted ice at a water droplet impact velocity of 52 m/s, which corresponds to





RPM in the AERTS facility. After the minimum thickness was accreted, the icing cloud was turned off and the rotor stand spun down. The same
ice removal technique as stated earlier was conducted to eliminate the last inch of ice from the tip, span-wise (region over the bus bar used to power
the heater). The rotor stand spun up to the desired RPM that corresponded to the centrifugal force at the heater zone being tested. When the matching
centrifugal force was achieved, power was delivered to the electro-thermal heaters at a selected power density. The maximum available power density
given the maximum power available for the heating system (100 kW), 0.385 W/cm² was the initial value before decreasing power density for the latter
test cases. The ice shedding times were quantified and recorded for the *Light* and *Medium* icing conditions. Once again, the L*ight* icing condition
pertains to a 0.2 g/m³ LWC and 20 µm MVD at a -8°C temperature. The M*edium* icing conditions are a 0.4 g/m³ LWC and 20 µm MVD at a -8°C
temperature. The parameters of the test matrix for the power variation experiments are shown in Table 4.

**Table 4:** Test matrix for de-icing IPS power variation.

| Heater | Span % | Heater Power Density (W/cm²) | Shedding RPM in AERTS | Icing Condition |
|---|---|---|---|---|
| 4 | 26.7 | 0.385, 0.33, 0.27, 0.225 | 286 | LIGHT & MEDIUM |
| 3 | 44.4 | 0.385, 0.33, 0.27, 0.225 | 369 | LIGHT & MEDIUM |
| 2 | 62.2 | 0.385, 0.33, 0.27, 0.225 | 437 | LIGHT & MEDIUM |

The test cases for the innermost heater zone, Heater 4, accreted an average of 6 mm of ice on the leading edge in *Light* icing conditions. For the
0.385, 0.33, 0.27 and 0.225 W/cm² power densities, the average shedding times were 28, 35, 83 and >180 seconds, respectively. The test cases for
Heater 4 on average accreted a 7 mm ice thickness on the leading edge in *Medium* icing conditions. For the 0.385, 0.33, 0.27 and 0.225 W/cm² power
densities, the average shedding times were 22, 42, 65 and >180 seconds, respectively. The data trend for the innermost heating section, zone 4, is shown
in Figure 24.

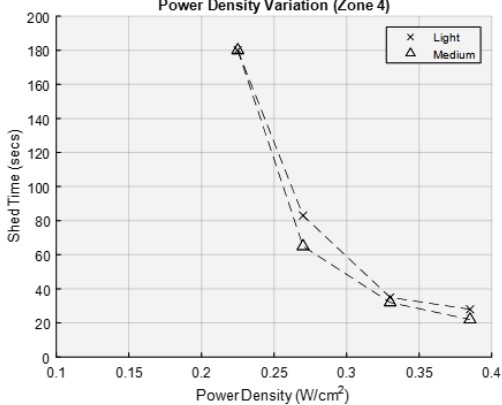

**Figure 24:** Power density variation shedding times for heater zone 4.
The test cases for heater zone 3 accreted an average ice thickness of 4 mm on the leading edge in *Light* icing conditions. For the 0.385, 0.33,
0.27 and 0.225 W/cm² power densities, the average shedding times were 25, 48, 88 and >180 seconds, respectively. Test cases in the *Medium* icing
conditions accumulated an average ice thickness of 5 mm. For the 0.385, 0.33, 0.27 and 0.225 W/cm² power densities, the average shedding times were
25, 38, 82 and >180 seconds, respectively. The data trend for heater zone 3 is shown in Figure 25.



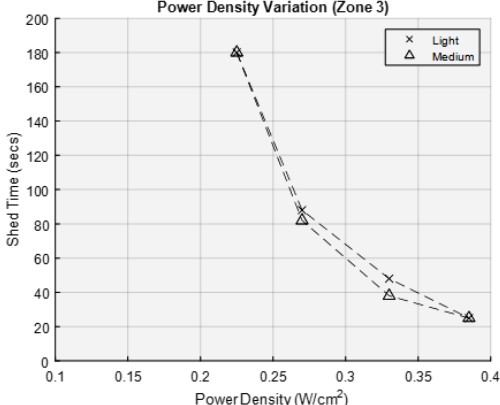

**Figure 25:** Power density variation shedding times for heater zone 3.

Lastly, heater zone 2 accreted 3 mm of ice on the leading edge for *Light* icing conditions. For the 0.385, 0.33, 0.27 and 0.225 W/cm$^2$ power densities, the average shedding times were 28, 47, 67 and >180 seconds, respectively. Test cases in the *Medium* icing conditions accreted an average ice thickness of 3.7 mm. For the 0.385, 0.33, 0.27 and 0.225 W/cm$^2$ power densities, the average shedding times were 24, 57, 85 and >180 seconds, respectively. The power variation data trend is presented in Figure 26.

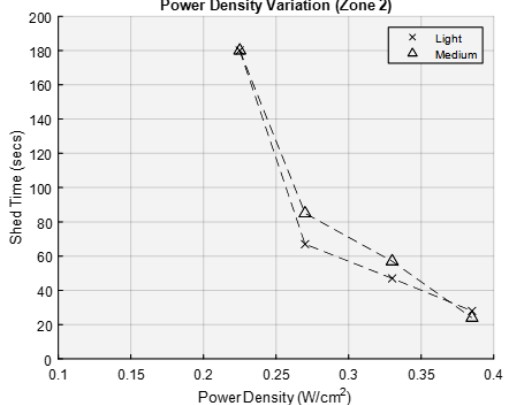

**Figure 26:** Power density variation shedding times for heater zone 2.

The data shows a parabolic trend indicating that the variation of power density plays a critical role in the heat transfer physics. Small changes in power density can result in shedding times above the "effective shedding" requirement. Therefore, the maximum power available on the 1.5 MW wind turbine established for the parametric design is needed for the de-icing ice protection system to perform in varying icing conditions given the heating zones selected.

## VI.    Controller Design

The controller is dependent upon the ice accretion rates, found in Table 2, and the minimum ice thickness required for effective shedding. A crucial hardware for the integration of the system is an ice detection sensor. It was not the aim of this research to explore the capabilities of suitable ice detection sensors. The controller interface needs to know when ice is initially present and the ice thickness after a selected time period. These two steps are needed for identifying the icing condition, since the ice accretion rate determines the icing condition experienced. Each icing condition has a repeatable time sequence operation to de-ice the rotor blades and minimize aerodynamic penalties. The time sequence incorporates the minimum ice thickness needed to ensure the ice accretion reaches that critical value to overcome cohesive forces. The controller time sequences for *Light*, *Medium* and *Severe* icing conditions are presented in Figures 27-29, respectively. The controller operations are illustrated as a flowchart in Figure 30.



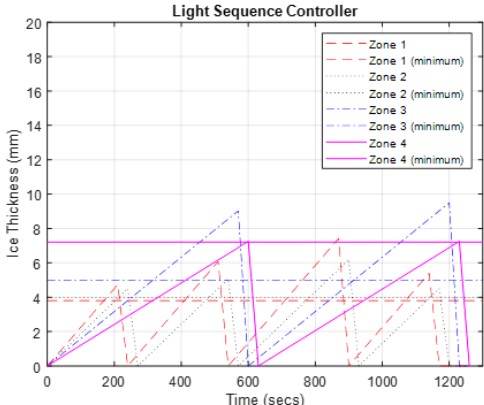

**Figure 27:** Repeatable time sequence for *Light* icing conditions.

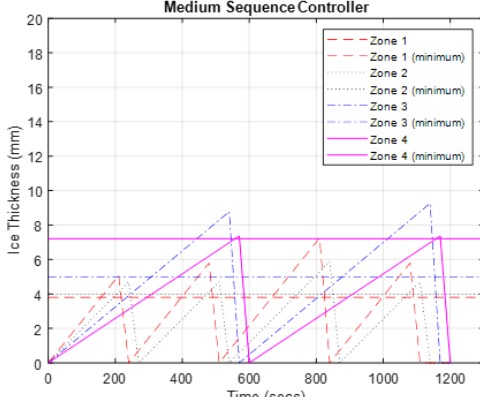

**Figure 28:** Repeatable time sequence for *Medium* icing conditions.

384        The horizontal lines represent the minimum ice thicknesses for each heater zone needed to produce cohesive failure. The slope of each line
represents the ice accretion rate (mm/second) found experimentally at each zone. When the ice thickness for each zone exceeds the critical thickness
value, the centrifugal force on the ice mass will dominate the ice cohesion force with the adjacent heater zone. The downward slope after the peak for
each zone represents the effective shedding requirement of 30 seconds.

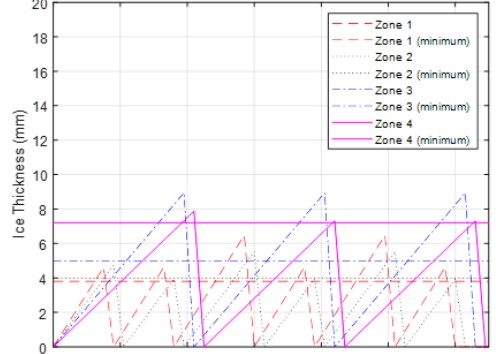




**Figure 29:** Repeatable time sequence for *Severe* icing conditions.
The most inboard zone, heater 4, has the smallest accretion rate and the longest time duration to reach the critical thickness value. Heater zone
3 must sacrifice aerodynamic performance to allocate the proper time to produce a shedding event for zone 4, because zone 3 must shed prior and be
ice-free before zone 4 is activated.
Knowing ice thickness rates ensures that the controller does not attempt to de-ice when the ice thickness has not reached a critical ice thickness.
If ice sensors are not available, a conservative approach could be to turn on the heaters for a time sequence corresponding to the Light icing condition,
which allows more time for ice to accrete. Such approach would introduce aerodynamic penalties for the Medium and Severe icing conditions, as the
ice thickness accreted will exceed the minimum thickness required for shedding.

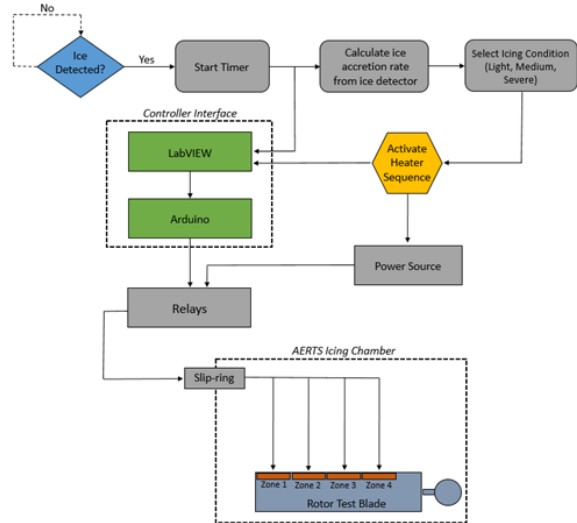

**Figure 30:** Flowchart of controller operations.
**VII.     Conclusions / Future Work**
The wind turbine industry is rapidly expanding and there is a need for an effective and low-power ice protection system. An electro-thermal
heater system was designed and tested for a representative 1.5 MW wind turbine. Experimental work was conducted to:
1.   Obtain the ice accretion slope along the span of the wind turbine blade utilizing the DU 93-W-210 airfoils for representative icing conditions.
2.   Determine the minimum thickness needed to effectively shed ice without cohesive bonding between heater zones, while delivering a 0.385 W/cm$^2$
power density to the electro-thermal heaters (maximum power density available on the turbine for the ice protection system).
3.   Explore the time variation of ice shedding by decreasing the power density to the electro-thermal heater to attempt to lower power delivery to the
zones.
4.   Design a time sequence controller based on ice accretion rates for varying icing conditions to minimize ice accretion mass and aerodynamic
penalties.

The ice accretion rates for the selected icing conditions ranged from 0.725 mm/min to 1.850 mm/min. Light icing conditions were created using
clouds at -8°C with a 0.2 g/m3 liquid water content (LWC) and water droplets of 20 µm median volumetric diameter (MVD). Medium icing condition
clouds had a LWC of 0.4 g/m3 and 20 µm MVD, also at -8°C. Severe icing conditions had an LWC of 0.9 g/m3 and 35 µm MVD at -8°C. The minimum
thicknesses for each heater zone beginning with the outboard section were 3.8 mm, 4.0 mm, 5.0 mm and 7.2 mm. These values were obtained
experimentally and represent the full-scale ice debonding of a heating zone, which overcomes cohesive bonding of adjacent ice regions. The accreted
ice thickness must reach these values at each heater zone to ensure the centrifugal force overcomes the ice cohesion force between zones. For the design
of the controller, an effective shedding time limit was imposed. With a 0.385 W/cm$^2$ heater power density, each zone was required to shed the minimum
ice thickness within 30 seconds. The power density was then varied to lower values to measure the time discrepancy between shedding events. It was
found that lower power densities are not suitable for the proposed configuration.
Accreting a certain thickness of ice to eliminate cohesive bonding, but also keeping the ice mass that shed to a minimum for the surrounding
safety, aerodynamic degradation minimization, and reduction of potential imbalance forces is critical. Understanding what type of weather the wind
turbine experiences at its location throughout the annual season is imperative, specifically, knowing the ice accretion rate at a given icing condition.
Current modeling tools provide a starting point to design such de-icing systems, but given the uncertainties of ice accretion to wind turbines, variations
on heater performance, and blade manufacturing schemes (affecting heat transfer), experimental testing procedures are suggested in this paper for the





design of de-icing systems for wind turbines. Future experiments should incorporate a test rig with multiple heater zones to effectively test for a full
system performance, including integration of an ice detection sensor.
The research was successful groundwork for the design of a de-icing ice protection system. Using ice thickness measurements allows for the
design of a time sequence algorithm for varying icing conditions and minimizing ice accretion mass. If the ice accretion rate per time is known, the
type of icing condition can be approximated and that appropriate heater sequence timing can be interpolated.

## VIII.    Acknowledgments

The work explored was funded by DuPont under a contract with The Pennsylvania State University Aerospace Department. The research was
directed with the advice and expertise of Professor, Dr. Jose Palacios. The collaboration between the teams was an amazing experience and
undoubtedly the reason for great results in this research.

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

Environments Conference 13 June 2016.