# Peer review of "19 20 21 22 23 24 25 26 27 28"

_Wind Energy Science, 2020_

## Referee Comment (RC1) · Anonymous Referee #1 · 30 Jun 2020

WES-2020-68

Design Procedures and Experimental Verification of an Electro-Thermal De-Icing System for Wind Turbines

By David Getz and Jose Palacios

The paper reports the design process of a de-icing system for HAWTs, based on an experimental analysis of an iced rotating DU 93-W-210 airfoil. At first, ice is accreted on the rotating model. Then the model is set to rotate at a precise speed, chosen to match the centrifugal force of a full-size blade at a certain radial position. The electro-thermal de-icing system is turned on, and, if the accreted ice is thicker than a certain threshold, the ice sheds due to centrifugal force within the maximum fixed time of 30 seconds.
This procedure is repeated for different radial positions. The minimum thickness required for ice shedding is determined and, at last, an effective time sequence is developed to shed portions of ice consecutively, from the tip to the inner part of the blade.

The English language is satisfactory. The topic is of sure interest for the WES reader.
The article seems to be a summary of much longer text(s) and, as such, contains some non-essential information as well as unnecessary repetitions. In some parts it also contains contradictory information. **Re-writing from scratch is advisable**.

The Reviewer has some major concerns about the experimental ice accretion procedure and its relationship with a real wind turbine, as follows:

> The representative 1.5MW wind turbine used as reference is too generic, and no information is provided about it, except for the supposed length of the blade (reported only in Figure 9). The rotational speed, which is used both for ice accretion experiments and to calculate the centrifugal force on the blade, should be provided explicitly (since it is a fundamental datum for the experiments), but can only be retrieved by putting together information from Table 1 and Figure 9. The resulting rotational speed should be 18rpm.

> The icing experiment is done on a single, scaled airfoil, but no information about scaling is provided within the article. In fact, scaling methods seem to be completely ignored. Reference [1] , included in the submission as Reference [32], is an entire Master Thesis about scaling methods developed at the very same AERTS Laboratory. In [2] the same author of [1] applies a scaling method to study a wind turbine blade at AERTS Laboratory.

> RPM is said to be matching "the impact velocity at a chosen span percentage of a wind turbine" (Line 226-227), but no information is provided about the free-stream velocity; inflow parameters seem to be neglected as well. Again, see [2] as reference.
> The authors also neglect the effect of chord and cross section (both of which vary along a blade span) on the collection efficiency of the airfoil (see, for instance, [3]).

> As a result, ice thickness is found to increase linearly with rotor span, "as expected" (line 236). The authors also feel confident to extrapolate the data trend outside the tested range (line 238). The result is surely valid for the experiment done and, more in general, the relation seems to be valid for a straight, untwisted, untapered (and non-lifting) blade, but is not valid for an actual wind turbine blade in power production. Some numerical results of ice accretion on a full blade can be found, for instance, in [4]. If this is a choice, in seek for simplicity, it should be pointed out.

Due to this lack of specificity, the time sequence found is strictly related to the experiment and provides qualitative data only. The design process is still valid. Again, if this is a choice, it should be pointed out.

Moreover, contradictory information is provided: lines 116-117 state "Representative icing conditions were selected and guided by FAR Appendix C icing conditions typical of aircraft environments", while lines 155-156 state "The density value was selected as it is representative of that seeing on wind turbines". In general, is not clear how these conditions were chosen.

In view of all this, the authors should (a) comment the choices made for ice accretion tests, by pointing out the reasons leading to these choices and the limits of validity of both the accretion test and the following results, or (b) repeat the experiment in more realistic conditions. In both cases the relationship with the considered wind turbine should be made clear and the set of boundary conditions should be justified properly.

[1] Han, Theoretical and Experimental Study of Scaling Methods for Rotor Blade Ice Accretion Testing," The Pennsylvania State University, 2011.
[2] Han et al., Scaled ice accretion experiments on a rotating wind turbine blade, J. Wind Eng. Ind. Aerodyn. 109 (2012) 55–67
[3] Homola et al., The relationship between chord length and rime icing on wind turbines, Wind Energ. 2010; 13:627–632
[4] Yirtici et al., Ice Accretion Prediction on Wind Turbines and Consequent Power Losses, J. Phys.: Conf. Ser. 753 (2016) 022022

Minor concerns are as follows:

- The authors state that the time sequence is designed to "minimize aerodynamic penalties" (line 377). In order to minimize aerodynamic penalties, ice at blade tip (zone 1 and, at most, zone 2) should be shed as soon as possible. Moreover, figures 27-29 can be difficult to read. Maybe a dashed line for the "minimum" line and a solid line for the others could be a better solution.

- the abstract is too long, detailed, redundant and confusing. It is advisable to make it shorter for better clarity.
  It also contains misleading information or information missing in the article, such as:

  ○ "Wind turbine representative **airfoils** […] were tested" (one single airfoil is tested);

  ○ "**The wind turbine sections were ½ scale models of the 80% span region of a generic 1.5 MW wind turbine blade.** A sample wind turbine configuration was selected to describe the design process. A 1.5 MW wind turbine was chosen". (Bold: information missing in the article. The rest is included to show the lack of consequentiality within the information provided by the abstract).

- some statements should require a reference, in particular:

    - Lines 83-85: "Typically, the LWC and MVD affect the thickness of the ice shape, while temperature and droplet impact velocity affect the surface roughness and adhesion strength of the ice."

    - Lines 92-93: "In general, the evaporative mode for anti-icing systems require about 5 times more energy to operate, rendering it as too expensive for wind turbines."

- There are a few errors to be corrected, as listed below:

    - Line 151: "Figure 6: Wind Turbine test blade representative (**Han, 2015**)".

    - Line 259 & Line 270: Table 2 & Table 3 should be corrected. Heater #, Span %, and data are mixed. Table 4 (line 348), on the contrary, is fine.

    - Lines 308-310: data provided are reversed. It should be "the minimum ice thicknesses were **6.8 mm** [text: 3.7 mm], 5 mm and **3.7 mm** [text: 6.8 mm], respectively." The same goes with all the data provided up to line 310.

    - Lines 458-467 the order of the References is wrong.

Other comments:

- Lines 62-63: "It is estimated that the capacity will reach 425 GW by 2015". This is meaningless in 2020. More in general, lines 55-69 give no added value to the article and could be omitted.

- The LEWICE study, which is cited but not presented, could be briefly introduced for completeness.

- More in general, if re-written, the article could contain more information in the same space, or the same information in less space, being shorter and clearer at the same time, and removing unnecessary repetitions and contradictions.

In view of the above comments, the Reviewer suggests that the paper undergoes a major revision, before being re-considered for publication on Wind Energy Science.

---

## Author Comment (AC1) · 14 Jul 2020

WES-2020-68

Dear Reviewer,
Thank you very much for taking the time to review and comment on our document. We were very impressed with the great details provided in the review comments and with the recommended modifications proposed. We will go over the document to address all the comments and suggestions.

Below, we have included details on how each comment provided will be addressed.
We would like to thank the reviewers again for all their time and effort. Their in-depth analysis and detailed comments have helped us improve the quality of the document.

Sincerely,
Jose Palacios

###########################################################################
Title: Design Procedures and Experimental Verification of an Electro-Thermal De-Icing System for Wind Turbines
By David Getz and Jose Palacios

Comments from reviewer:

The paper reports the design process of a de-icing system for HAWTs, based on an experimental analysis of an iced rotating DU 93-W-210 airfoil. At first, ice is accreted on the rotating model. Then the model is set to rotate at a precise speed, chosen to match the centrifugal force of a full-size blade at a certain radial position. The electro-thermal de-icing system is turned on, and, if the accreted ice is thicker than a certain threshold, the ice sheds due to centrifugal force within the maximum fixed time of 30 seconds.

This procedure is repeated for different radial positions. The minimum thickness required for ice shedding is determined and, at last, an effective time sequence is developed to shed portions of ice consecutively, from the tip to the inner part of the blade.

The English language is satisfactory. The topic is of sure interest for the WES reader.
The article seems to be a summary of much longer text(s) and, as such, contains some non-essential information as well as unnecessary repetitions. In some parts it also contains contradictory information. Re-writing from scratch is advisable.

*Answer from author in italics:*
*Thank you for your comment on the repeated text. We will re-visit the document to eliminate any repetitions and eliminate unnecessary details.*

The Reviewer has some major concerns about the experimental ice accretion procedure and its relationship with a real wind turbine, as follows:

The representative 1.5MW wind turbine used as reference is too generic, and no information is provided about it, except for the supposed length of the blade (reported only in Figure 9).

*Information on the selection of the wind turbine representative airfoil is available on the Journal published by Blasco et al. in 2015. Detailed aerodynamic analysis was conducted in the design on the selected airfoil. These details will be summarized in the paper and the journal publication (instead of the MS thesis) will be referenced.*

The rotational speed, which is used both for ice accretion experiments and to calculate the centrifugal force on the blade, should be provided explicitly (since it is a fundamental datum for the experiments),

but can only be retrieved by putting together information from Table 1 and Figure 9. The resulting rotational speed should be 18rpm.

*Some of the information used in the work (such as operation conditions of the wind turbine modeled with generic airfoil shapes and conditions) was deemed proprietary by the manufacturer and operators, but the RPM is indeed 18. We will add this information explicitly in the document.*

The icing experiment is done on a single, scaled airfoil, but no information about scaling is provided within the article. In fact, scaling methods seem to be completely ignored. Reference [1] , included in the submission as Reference [32], is an entire Master Thesis about scaling methods developed at the very same AERTS Laboratory. In [2] the same author of [1] applies a scaling method to study a wind turbine blade at AERTS Laboratory.

*The referenced thesis will be replaced with correspondent journal papers that summarize the approaches. A brief description of the ice scaling laws applied will be added to the paper.*

RPM is said to be matching "the impact velocity at a chosen span percentage of a wind turbine" (Line 226-227), but no information is provided about the free-stream velocity; inflow parameters seem to be neglected as well. Again, see [2] as reference. The authors also neglect the effect of chord and cross section (both of which vary along a blade span) on the collection efficiency of the airfoil (see, for instance, [3]).

*Information regarding free-stream velocity at the effective span sections analyzed will be explicitly added to the paper. The reviewer is correct that tapering and twist of production blades was ignored on the effort. The research explains the design procedure to design electrothermal ice protection systems for wind turbines, but it is limited to straight, un-twisted blades at 0 degrees AoA. The design procedure would be applicable to varying airfoil shapes, AOA and chord for production blades, but pertinent ice scaling laws (for experimental testing) and/or modeling of ice accretion for these span locations would need to be conducted. The authors will add these statements to the document.*

As a result, ice thickness is found to increase linearly with rotor span, "as expected" (line 236). The authors also feel confident to extrapolate the data trend outside the tested range (line 238). The result is surely valid for the experiment done and, more in general, the relation seems to be valid for a straight, untwisted, untapered (and non-lifting) blade, but is not valid for an actual wind turbine blade in power production. Some numerical results of ice accretion on a full blade can be found, for instance, in [4]. If this is a choice, in seek for simplicity, it should be pointed out.

*The authors will emphasize the idealized blades conditions selected for simplicity, as the goal of the work is to develop a design procedure for the ice protection system. The procedure presented could be applied in the future by industry designers to twisted, tapered, lifting blades with updated models or acquired ice accretion thickness to in-production, varying span wise blade geometries.*

Due to this lack of specificity, the time sequence found is strictly related to the experiment and provides qualitative data only. The design process is still valid. Again, if this is a choice, it should be pointed out.

*Again, the author is correct. It was not the intent of the authors to design a system for a potentially proprietary blade geometry, but to put forth a design process for an electrothermal de-icing system for wind turbines. A simplified, straight, untwisted blades at 0 degrees AoA was used.*

Moreover, contradictory information is provided: lines 116-117 state "Representative icing conditions were selected and guided by FAR Appendix C icing conditions typical of aircraft environments", while lines 155-156 state "The density value was selected as it is representative of that seeing on wind turbines". In general, is not clear how these conditions were chosen.

*The reviewer is correct that the authors should clarify the icing conditions selected. Guidance from*

*wind turbine operators was followed, indicating that glaze conditions were of interest as the ice shapes accreted in AERTS were representative of those seen in the field for a specific wind turbine farm. Glaze conditions were then selected form Appendix C envelopes. The authors will explain the selection process for the icing conditions in the paper.*

In view of all this, the authors should (a) comment the choices made for ice accretion tests, by pointing out the reasons leading to these choices and the limits of validity of both the accretion test and the following results, or (b) repeat the experiment in more realistic conditions. In both cases the relationship with the considered wind turbine should be made clear and the set of boundary conditions should be justified properly.

*The authors will edit the paper to avoid repetitions and will include the explanations to the reviewer's comments. Thank you very much for taking the time to review our work and for your interest on the topic.*

[1] Han, Theoretical and Experimental Study of Scaling Methods for Rotor Blade Ice Accretion Testing," The Pennsylvania State University, 2011.
[2] Han et al., Scaled ice accretion experiments on a rotating wind turbine blade, J. Wind Eng. Ind. Aerodyn. 109 (2012) 55–67
[3] Homola et al., The relationship between chord length and rime icing on wind turbines, Wind Energ. 2010; 13:627–632
[4] Yirtici et al., Ice Accretion Prediction on Wind Turbines and Consequent Power Losses, J. Phys.: Conf. Ser. 753 (2016) 022022

Minor concerns are as follows:

- The authors state that the time sequence is designed to "minimize aerodynamic penalties" (line 377). In order to minimize aerodynamic penalties, ice at blade tip (zone 1 and, at most, zone 2) should be shed as soon as possible. Moreover, figures 27-29 can be difficult to read. Maybe a dashed line for the "minimum" line and a solid line for the others could be a better solution.

    *The images will be redone with bigger fonts and thicker lines*

- the abstract is too long, detailed, redundant and confusing. It is advisable to make it shorter for better clarity.
  It also contains misleading information or information missing in the article, such as:

    - "Wind turbine representative **airfoils** […] were tested" (one single airfoil is tested);

    - "**The wind turbine sections were ½ scale models of the 80% span region of a generic 1.5 MW wind turbine blade.** A sample wind turbine configuration was selected to describe the design process. A 1.5 MW wind turbine was chosen". (Bold: information missing in the article. The rest is included to show the lack of consequentiality within the information provided by the abstract).

        *The information can be found in Blasco et al. 2015, but the authors agree that a description of the airfoil selection should be added to the document. We will review the entire document for redundancies including the abstract*

- some statements should require a reference, in particular:

    - Lines 83-85: "Typically, the LWC and MVD affect the thickness of the ice shape, while temperature and droplet impact velocity affect the surface roughness and adhesion strength of the ice."

*A reference will be added.*

- o Lines 92-93: "In general, the evaporative mode for anti-icing systems require about 5 times more energy to operate, rendering it as too expensive for wind turbines."

  *A reference will be added.*

- There are a few errors to be corrected, as listed below:

  - o Line 151: "Figure 6: Wind Turbine test blade representative (**Han, 2015**)".

  - o Line 259 & Line 270: Table 2 & Table 3 should be corrected. Heater #, Span %, and data are mixed. Table 4 (line 348), on the contrary, is fine.

    *Thank you for pointing out this typo. It will be addressed on the edited document.*

  - o Lines 308-310: data provided are reversed. It should be "the minimum ice thicknesses were **6.8 mm** [text: 3.7 mm], 5 mm and **3.7 mm** [text: 6.8 mm], respectively." The same goes with all the data provided up to line 310.

    *Thank you for pointing out this typo. It will be addressed on the edited document.*

  - o Lines 458-467 the order of the References is wrong.

    *Thank you for pointing out this typo. It will be addressed on the edited document.*

Other comments:

- Lines 62-63: "It is estimated that the capacity will reach 425 GW by 2015". This is meaningless in 2020. More in general, lines 55-69 give no added value to the article and could be omitted.

- The LEWICE study, which is cited but not presented, could be briefly introduced for completeness.

- More in general, if re-written, the article could contain more information in the same space, or the same information in less space, being shorter and clearer at the same time, and removing unnecessary repetitions and contradictions.

In view of the above comments, the Reviewer suggests that the paper undergoes a major revision, before being re-considered for publication on Wind Energy Science.

*Thank you again for your time and comments.*

---

## Referee Comment (RC2) · Anonymous Referee #2 · 3 Mar 2021

The article wes-2020-68 by Getz and Palacios looks complete and should be accepted for publication. The contents are thoroughly and clearly presented.

---

## Author Response (AR1)

Dear Alessandro,

We are thrilled to have our work move on to the discussion state. As requested, we have improved the quality of the figures and reviewed the referring to match template requirements.

Looking forward to continuing to work with your review team to have a chance to share or work with the community.

Sincerely,

Prof. Palacios

---

## Author Response (AR2)

Dear reviewers and editor,

Thank you again for taking the time to review our work and provide detailed comments to reinforce our document disseminating our findings. Please see below specific answers to the reviewer's comments in blue. Our team really appreciate your efforts.

Sincerely,

Prof. Palacios

**Suggestions for revision or reasons for rejection (will be published if the paper is accepted for final publication)**
The paper describes the design process of an electro-thermal de-icing system for HAWTs. The procedure is based on experiments of ice accretion and ice shedding due to centrifugal force on a rotating blade section. Once data is obtained at different rotational speeds, corresponding to different radial positions, a control sequence is proposed for de-icing a full blade in different icing conditions.

The design procedure proposed is interesting, and the topic is of sure interest to the reader of Wind Energy Science.

The English language is not fully satisfactory. Although its usage is globally correct, some typos and grammatical errors should be corrected (e.g., in ll. 19-20, 39, 45, 55, 57, 68, 122, 153, 159, 185, etc.). Providing some examples from Section I, I advise as well:
1. to check for consistency in the flow of information for easier reading (e.g.: ll. 55-65 regard wind energy in general; ll. 68-73 regard issues related with icing on wind turbines; ll. 56-58 regard wind energy in cold regions, which might be moved right before l. 68);
Thank you for your recommendations and for pointing out the typos in the paper. We have read over the document and we have addressed mentioned typos.
2. to avoid wordy repetitions (e.g., in ll. 82-87, the subject «de-icing» is repeated in every sentence);
We believe that making clear the type of system being discussed (de-icing vs anti-icing) is critical to avoid confusion, even if it requires labeling the system in every sentence. We have reduced the number of mentions while hopefully avoiding confusion.
3. to omit unnecessary information, or better contextualize it (e.g., ll. 91-96, which include issues (1) and (2) as well).

The abstract should be rewritten to be more generic and include fewer details: it is difficult to understand all the information provided before reading the whole paper.
Information related to the specific icing conditions tested was removed from the abstract.
An overview of the structure of the paper and its contents is missing. I suggest including it at the end of Section I to facilitate reading.
The objectives section was re-written to enumerate the tasks which are followed in structure of the proposed paper and that describes the design process

I have one major concern about the methodology applied.
In particular, it is not clear why the cross-sectional area of accreted ice on the scaled model should match the one of the full wind turbine, as per equations (1) and (2) (ll. 171-172). This assumption, which was not justified and cannot be taken for granted a priori, is critical within the design process since it assigns a centrifugal force that may not be linked to the full-scale model. In fact, the ice accretion rate depends on the shape and the size of the object as well [1, 2, 3].
Moreover, the experiments are carried out on a «1/2 scale model of the 80% span region of a generic 1.5 MW wind turbine blade». It is stated that «representative ice shapes were first accreted as per ice scaling laws provided by Bond et al., 2004» (ll. 156-157); it is not clear if (and which) scaling law is applied, and if this is related to the assumption above. Later, it is stated that the impact velocity only was matched during ice accretion experiments (ll. 232-234). The authors should clarify the approach chosen.
Since the design process is intended for a full-scale model, I invite the authors to discuss the

legitimacy or the consequences of their assumptions, either numerically or by providing results from existing literature.

A "Testing Methodology" section was added to the document to clarify the need for representative types of ice and accretion areas. It is critical to match the type of ice accreted on the ½ scale airfoil to that of a full-scale, since the cohesion strength of ice is dependent on the type of ice. Cohesion between heated zones must be overcome during the shedding process.

My second concern regards unclear, confusing, or contradictory information provided within the paper. In particular:
1. Linked to what already written above, confusing information is provided regarding the influence of different parameters on ice accretion. In ll. 74-75, authors should mention that the shape and size of the object affect the ice accretion rate on the object itself [1, 2, 3]. In ll. 79-81, it is implicitly stated that impact velocity does not affect the thickness, which is certainly wrong and contradicts the results of the experiments. More attention should be paid while defining the effect of the different parameters on ice accretion.

Impact velocity does affect the ice accretion rate. It has been made clear that in the effort the ice accretion rate recorded is used in the heater controller design procedure, but ultimately, the ice accretion rate must be measured real time in the field by ice thickness sensors.

2. It is not clear if experiments for blade span r/R = 0.8 were run. Line 239 states that «80% span was not tested in the AERTS facility due to the potential rotor imbalance on the test stand at such RPM» (Table 1 still reports «3, 5 and 7» mins of accretion time for that blade span); accordingly, results for that blade span are obtained through extrapolation (please note that the order of results reported in Table 2 may be wrong). In the following Section (V.B), it seems that experiments at r/R = 0.8 will be performed; however, results in Figure 20 do not include r/R = 0.8, so it is not clear how results in Figure 23 are obtained for that blade span. Once more, in Section V.C, r/R = 0.8 is not tested and, accordingly, results are not presented. Was the case of r/R = 0.8 ever tested?

80% span was not tested due to rotor imbalance concerns upon shedding. It has been made clear on the text that tests did not include 80% span testing conditions.

Minor concerns regard occasional scarce attention to details. Some of them are listed below.
- Data described in line 62 does not match what is shown in Figure 2. See modification on text
- «currently», in line 63, refers to 2011 data. In 2020, it produced 8.42% of electricity in the US [4].
Thank you very much for the additional information. We have added it to the text.
- Units of measure should be written consistently: e.g., 10_GW (line 62), g/m^3 (ll. 226-227).
- Some acronyms have never been introduced (IRT: Icing Research Tunnel; FAR: Federal Aviation Regulations). The acronyms have been fully written.
- In Figure 11, some lines are overlapping text boxes. Fixed
- Resolution of Figure 30 should be improved. Redone
- I suggest changing the title of Section VII to "Conclusions and Future Work".
- The author of NASA/CR—2004-212875 is Anderson only. Correct.
- The Bibliography should be ordered alphabetically. Done

Given the comments above, I suggest that the paper undergoes a major revision, before being reconsidered for publication on Wind Energy Science.

**Suggestions for revision or reasons for rejection (will be published if the paper is accepted for final publication)**
Dear Authors,
thank you for addressing most concerns on the submitted paper, which can now be accepted for publication. Congratulations!

I only have a short suggestion: please comment that Appendix C conditions are taken from FAA regulations and their applicability to wind turbine icing is not straightforward. I expect that in-flight icing conditions differ from ground level ones, where I am expecting operating parameters to be different as well as a strong influence of the terrestrial boundary layer.

Thank you. We have written on the document that the types of ice selected are representative of aircraft icing and not wind turbines, since a comprehensive icing envelope for wind turbines are not available to our knowledge. Ultimately it is not the goal of the paper to match ice shapes for a given wind turbine but showcase the proposed design process for a wind turbine ice protection system, with emphasis on the need to overcome cohesive forces between adjacent heater zones.